# Effects of Zeolite as a Drug Delivery System on Cancer Therapy: A Systematic Review

**DOI:** 10.3390/molecules26206196

**Published:** 2021-10-14

**Authors:** Jessica Hao, Ivana Stavljenić Milašin, Zeynep Batu Eken, Marinka Mravak-Stipetic, Krešimir Pavelić, Fusun Ozer

**Affiliations:** 1Department of Biology, School of Arts and Sciences, University of Pennsylvania, Philadelphia, PA 19104, USA; haoje@sas.upenn.edu; 2Dental Polyclinic Zagreb, 10000 Zagreb, Croatia; frckava@gmail.com; 3Department of Restorative Dentistry, Yeditepe University, 34728 Istanbul, Turkey; zeynep.eken@yeditepe.edu.tr; 4Clinical Department of Oral Medicine, School of Dental Medicine, University of Zagreb, 10000 Zagreb, Croatia; mravak@sfzg.hr; 5Faculty of Medicine, Juraj Dobrila University of Pula, HR-52100 Pula, Croatia; pavelic@unipu.hr; 6Department of Preventative and Restorative Sciences, School of Dental Medicine, University of Pennsylvania, Philadelphia, PA 19104, USA

**Keywords:** zeolite, clinoptilolite, zeolitic imidazolate framework, metal-organic framework, cancer therapy, nanocarrier, anticancer drug, antineoplastic drug, drug delivery system, systematic review

## Abstract

Zeolites and zeolitic imidazolate frameworks (ZIFs) are widely studied as drug carrying nanoplatforms to enhance the specificity and efficacy of traditional anticancer drugs. At present, there is no other systematic review that assesses the potency of zeolites/ZIFs as anticancer drug carriers. Due to the porous nature and inherent pH-sensitive properties of zeolites/ZIFs, the compounds can entrap and selectively release anticancer drugs into the acidic tumor microenvironment. Therefore, it is valuable to provide a comprehensive overview of available evidence on the topic to identify the benefits of the compound as well as potential gaps in knowledge. The purpose of this study was to evaluate the potential therapeutic applications of zeolites/ZIFs as drug delivery systems delivering doxorubicin (DOX), 5-fluorouracil (5-FU), curcumin, cisplatin, and miR-34a. Following PRISMA guidelines, an exhaustive search of PubMed, Scopus, Embase, and Web of Science was conducted. No language or time limitations were used up to 25th August 2021. Only full text articles were selected that pertained to the usage of zeolites/ZIFs in delivering anticancer drugs. Initially, 1279 studies were identified, of which 572 duplicate records were excluded. After screening for the title, abstract, and full texts, 53 articles remained and were included in the qualitative synthesis. An Inter-Rater Reliability (IRR) test, which included a percent user agreement and reliability percent, was conducted for the 53 articles. The included studies suggest that anticancer drug-incorporated zeolites/ZIFs can be used as alternative treatment options to enhance the efficacy of cancer treatment by mitigating the drawbacks of drugs under conventional treatment.

## 1. Introduction

Zeolites are minerals with a tetrahedral crystalline structure formed by dense networks of AlO_4_ and SiO_4_ sharing oxygen atoms [1,2,3]. These aluminosilicate networks create regularly distributed micropores and cavities that range between 4–12 Å in size [4]. The pores and cavities can exchange water, ions, and polar molecules with the surrounding environment, giving zeolites unique ion exchange properties and absorption capacities [2]. Absorption may occur on both the outer and inner surfaces of the material and are governed by the ability of molecules to fit into the micropores [4].

Zeolites were first discovered in the 18th century by Swedish mineralogist Axel Fredrik Crønsted, who introduced the term “zeolite” from the Greek ζέω (zéō), meaning “to boil”, and λίθος (líthos), meaning “stone”, owing to their natural properties he described [4]. Nowadays, zeolites have a wide range of industrial, environmental, and biomedical applications [1,2,3,4,5] and are available as both natural minerals and artificially produced materials. Of the over 40 natural and 230 synthetic zeolites known [4,6], many are subjected to diverse applications that harness the porous characteristic, ion exchange property, and high adsorption capacity of zeolites [3,4,6]. In addition, zeolites possess channels and/or cavities linked by channels, which give it a unique structural feature over other aluminosilicate and crystalline materials.

In recent years, the special properties of zeolites have garnered much attention towards the material in biomedicine [4,6]. Naturally occurring zeolite clinoptilolite is the most widely used zeolite in various biomedical applications [7] and for environmental and alimentary decontamination from radioactivity and toxic substances, respectively [2,4]. Clinoptilolite is currently the only zeolite registered in the EU as a medical device and can be used in oral treatment. Zeolite clinoptilolite can be also used as drug carriers and delivery systems. One potentially excitatory pharmacological application of zeolites and mesoporous silicates could be the encapsulation of different ions and molecules (i.e., proteins) with delayed-release properties. Zeolites are also used as support materials for enzymes and antibodies, and solubilization of drugs by zeolite-surfactant complexes may lead to new uses such as drug delivery systems and controlled-release chemicals. Furthermore, the use of microcapsules containing an enzyme-zeolite preparation may be a potential route to urea removal. Finally, commercial zeolites can act as a slow-release carriers for a number of drugs, particularly anthelmintics. It has been suggested that the slow release of drugs from the zeolite matrix may improve the drugs’ efficiency [8].

Although zeolites hold great potential for clinical use [1,2,4,7,9], it should be noted that several subtypes have inherent cytotoxic properties that cause detrimental effects toward the human body [2,4]. One of the most well-known toxic zeolites is erionite, a type of naturally occurring fibrous zeolite, which behaves like asbestos by causing lung cancer and malignant mesothelioma [10,11,12]. In addition, other types of fibrous zeolites, such as scolecite and offretite, may modify cellular organization and mitochondrial function [13]. Generally, it has been found that the size, shape, and composition of zeolitic particles have a strong effect on zeolite toxicity. For example, although microsized zeolite Linde Type L (LTL) and Linde Type A (LTA) show minimal cytotoxicity, the toxic effects of their nanosized counterparts vary greatly due to the crystal shape and alumina component of the zeolites. Pure silica nanosized LTL and LTA with a spherical morphology have no toxicity, while alumina containing LTL and LTA are toxic [14,15]. Recent toxicity studies have demonstrated that the internal surface of the zeolites does not affect the toxicity of the material since it does not interact with the surrounding biological environment [16,17]. Therefore, when considering the type of zeolite to be implemented into biomedical applications, it is important to differentiate certain types of zeolites with known toxic and carcinogenic effects from other types of zeolites (i.e., clinoptilolite) with therapeutic, anti-inflammatory, antiproliferative, and pro-apoptotic properties.

Zeolites are classified based on their pore structure, pore size, and chemical composition of silica and aluminum [18]. Specifically, nano-sized zeolitic particles have great potential in therapeutic drug delivery, with mesoporous silica-based nanoparticles as one of the most popular materials investigated in drug release research [19,20] (Table 1). Despite the advancements made in anticancer research in recent decades, conventional anticancer drugs still face drawbacks, such as poor specificity and high toxicity, which can lead to systemic toxicity for healthy tissues [21,22]. Using drug delivery systems as a new strategy to administer pre-existing therapeutic compounds can resolve these drawbacks since they can selectively release the drug in cancer cells while minimally impacting normal cells. An effective therapeutic drug carrier should meet the following five criteria: (1) be able to carry drugs with a high loading amount, (2) demonstrate the ability to steadily release drugs in a prolonged manner, (3) possess an easily-engineerable surface for the facile control of its in vivo fate, (4) be easily detectable by imaging techniques, and (5) undergo biotransformation and excretion with minimal side effects [23,24]. Unfortunately, conventional drug delivery systems today do not fulfil the listed criteria, and there is a need to search for a potential new carrier of anticancer therapeutics. Recently, the ability of zeolites to deliver various biomedical drugs have increased the material’s popularity in research to improve anticancer therapy [18]. Research regarding the use of zeolitic drug delivery systems in place of traditional drug delivery methods reveal better cellular uptake and less side effects without diminishing the intended pharmacological effects. It is also anticipated that further research on the surface modification of zeolites will be expanded in the future for their use in cancer therapy as drug delivery agents [14,25].

In addition to zeolites, metal-organic frameworks (MOFs) have a variety of advantages over other drug delivery systems due to their defined crystalline framework and flexibility in the combination of both organic and inorganic compounds [26,27,28,29]. Amongst the various types of MOFs, zeolitic imidazolate frameworks (ZIFs) are most frequently studied as drug delivery systems due to their biocompatibility at low concentrations [30,31,32], relative ease of synthesis [32,33,34,35,36], and pH-response [25,37]. ZIFs have especially garnered interest as pH-sensitive drug carriers due to their high drug loading capabilities and biodegradability [38]. Although ZIFs remain stable in water and aqueous NaOH, their framework quickly disintegrates in acidic solutions [39,40]. This indicates that the pH sensitivity of ZIFs can aid in the development of ZIF-based drug-release platforms [39]. Conveniently, cancer cells possess more acidic microenvironments compared with normal cells. This difference can be harnessed to achieve tumor-specific recognition and target treatment through anticancer drug-incorporated ZIF nanoplatforms [22]. Specifically, ZIF-8, which is constructed by the copolymerization of Zn with 2-methylimidazole, has been frequently used in studies regarding the delivery of DNA, protein, and drugs [41].

Recently, there has been increased interest and a multitude of studies regarding the potential of zeolites and ZIFs in enhancing the efficacy of pre-existing anticancer drugs. However, to the best of our knowledge, there is no systematic review that assesses and synthesizes the outcomes of these studies. The present systematic review focuses on existing literature regarding the therapeutic effects of zeolites and ZIFs as carriers of anticancer drugs such as doxorubicin (DOX), 5-fluorouracil (5-FU), curcumin, cisplatin, and miR-34a. Therefore, this paper delivers a novel and comprehensive overview of available evidence on the topic. Additionally, the results reported in the paper may identify gaps in knowledge within the field that can potentially guide the direction of future research.

The aim of this systematic study was to review both in vitro and in vivo studies that evaluated the potential enhancement of various anticancer drugs using zeolites/ZIFs as a drug delivery nanoplatform. If results are found to be promising, potential future directions may include clinical trials that may ultimately lead to the commercial application of the drug carrier. In addition, it would be valuable to investigate the drug delivery system in a variety of cancers through future studies to discern if there is a greater efficacy in certain types of cancers over others.

## 2. Discussion

In the present systematic review, in vivo studies as well as in vitro experiments on cytotoxicity and drug-release were examined based on the anticancer drug loaded within zeolites/ZIFs. Through the literature search, it was determined that there is great heterogeneity in the experimental methods and conditions between studies reporting on zeolites/ZIFs as anticancer drug delivery systems. The most common anticancer drug loaded were doxorubicin (DOX) [21,22,25,32,37,38,40,42,43,44,45,46,47,48,49,50,51,52,53,54,55,56,57,58], 5-fluorouracil (5-FU) [59,60,61,62,63,64,65,66,67,68], curcumin [14,69,70], cisplatin [71,72], and miR-34a [73,74]. The articles also employed a variety of both ZIFs and natural/synthetic zeolites (Table 1). Of the 53 articles included in this study, 36 studies utilized ZIFs [22,25,30,32,37,38,40,42,43,44,47,48,51,52,53,54,55,56,57,58,59,60,61,64,66,70,71,72,74,75,76,77,78,79,80,81], 9 studies utilized faujasite (FAU) [16,50,62,64,67,68,75,82,83], 5 studies utilized Zeolite A/Linde Type A (LTA) [14,49,63,68,82], and 3 studies utilized clinoptilolite [53,84,85].

### 2.1. Doxorubicin (DOX)

Doxorubicin (DOX), an anthracycline antibiotic with antineoplastic activity, is frequently used as an anticancer drug in chemotherapy. DOX acts on the human body by intercalating itself between the base pairs of the DNA double helix [86]. This inhibits topoisomerase II activity, prevents DNA replication, and ultimately hinders protein synthesis [47,86]. Despite its prevalence, however, DOX lacks the ability to target certain areas of the body and possesses strong cardiotoxic effects [32]. Therefore, a more effective delivery system is required to mitigate the side effects of DOX on the body.

#### 2.1.1. The Effects of the pH-Sensitive ZIF-8 and Zeolites on DOX Release

Studies relating to the DOX-releasing properties of ZIF-8 are diverse in nature, but they share a commonality in that the high sensitivity of ZIF-8 to pH makes it possible to accurately release anticancer drugs in the acidic tumor microenvironment [25]. In the study of Tan et al. DOX was loaded into MnO_2_@ZIF-8 and utilized a CCK-8 assay to determine that DOX/MnO_2_@ZIF-8 can significantly reduce the viability of LLC cells. In addition, mice were subjected to DOX/MnO_2_@ZIF-8 and demonstrated a reduction of tumor volume and increase of apoptotic cells. Therefore, MnO_2_@ZIF-8 may serve as a nanocarrier of DOX in lung cancer treatment [37]. Sharsheeva et al. combined ZIF-8 with a semiconductor photocatalytic agent that induces a local pH gradient in response to external electromagnetic radiation. This system was found to release DOX in a quantity that successfully suppresses neuroblastoma cells [44]. Kang et al. encapsulated copper bismuth sulfide nanoparticles and rare-earth down-conversion nanoparticles into ZIF-8 before loading the platform with DOX. The encapsulated components were released in response to a change in pH, and a moderate dose inhibited 87.6% of tumors with x-ray irradiation [43]. Li et al. engineered silk sericin into ZIF-8 to overcome poor circulation stability and unexpected drug leakage into blood circulation, both issues that may limit the benefits of chemotherapy. The synthesized nanoplatform has tumor-specific biodegradability induced by the low pH environment, efficient drug uptake, and substantial tumor permeability effects [25]. Moreover, Yan et al. sought to overcome a drawback of ZIF-8, which has low affinity to non-electron rich drugs and lacks surface functional groups. The study modified DOX with a pH-sensitive linker containing two carboxyl groups, which can anchor itself to the ZIF-8 surface to form a prodrug. In an acidic tumor environment, the pH-sensitive linker is cleaved, which yields inherent benefits of more precisely controlling the release of DOX [32]. Finally, Lei et al. grew ZIF-8 on the surface of micelles to form a core-shell nanocomposite. The inner cavity of the micelle acted as a DOX-hydrochloride storage reservoir, while the outer ZIF-8 coating acted as a pH-controlled gatekeeper of drug release. The core-shell nanocomposite can not only be successfully internalized by cancer cells to release DOX under the acidic intracellular environment but can also present lower cytotoxicity compared to free DOX towards normal cells [22].

In addition to harnessing the pH sensitive properties of ZIF-8, ZSM-5, a type of zeolite, can also be used as pH-responsive drug delivery systems. One study fabricated hollow mesoporous ellipsoids with ZSM-5 and chitosan that can load DOX at a 95.8% loading efficiency. In in vitro experiments with healthy blood and tissue-simulating media, the ellipsoids slowly released DOX to the surrounding environment. In contrast, in tumor cells, the ellipsoids rapidly released DOX, which resulted in the considerable apoptosis of MG63 cancer cells [46]. ZSM-5 and chitosan were also combined to form nanodisks, which demonstrated a greater DOX loading efficiency of 97.7%. The nanodisk drug carriers efficiently inhibited tumors with minor side effects, especially in cardiac toxicity [21]. Therefore, both ZIF-8 and ZSM-5 can efficiently be used as pH-sensitive drug carriers to enhance the specificity of DOX release.

#### 2.1.2. Dual Stimuli to Enhance DOX Release

Apart from solely relying on changes in pH for drug release, studies have utilized dual stimuli to further increase the specificity of the therapeutic platform [38,49,52,57,58]. Wu et al. explored this unique property of ZIF-8 by synthesizing a pH-responsive nanoplatform that integrated polydopamine, which greatly increased the biocompatibility of ZIF-8 in cytotoxicity and in vivo acute toxicity evaluations. Under the dual stimuli of a near-infrared (NIR) laser and an acidic environment, the DOX release rate increased from 21% to 78% [38]. A similar study exploring the combined effects of pH and a NIR laser constructed ZIF-8 Janus nanoparticles with lactobionic acid-gold nanorods on CT image-guided synergistic chemo-photothermal theranostics. The unique method not only had numerous advantages in cancer imaging, but also significantly inhibited the tumors in vivo by releasing pre-loaded DOX [57]. NIR laser stimuli exhibit promising results not only with pH change, but also with ultrasound stimulation. The multimodal therapy allowed DOX@LTA zeolites to increase its therapeutic efficiency in the deep sites of tumors [49]. Overall, these three studies reveal that the dual integration of a NIR laser and other stimuli allows both zeolites and ZIFs to serve as effective DOX-releasing platforms.

Dual stimulation drug release was also investigated by combining the stimulatory effects of low pH and high levels of glutathione, a compound that is present in high concentrations in tumor cell microenvironments. In this study, molecularly imprinted polymer (MIP)-stabilized fluorescent ZIF-8 was more likely phagocytosed and more lethal to MCF-7 breast cancer cells compared to other cells and nanoparticles. In addition, MIP-stabilized fluorescent ZIF-8 had the best inhibitory effect on the growth of MCF-7 tumors in mice [58]. He et al. examined the control of light and pH on the DOX hydrochloride-releasing properties of Au@ZIF-8. This study demonstrated especially promising results, showing that Au@ZIF-8 with only 10 μM of DOX hydrochloride can result in 98% HeLa cell-killing activity after 30 min of light irradiation [52]. Overall, ZIF-8 as a drug delivery platform can be induced by a variety of stimuli, which further supports promising applications of the nanoplatform in cancer therapeutic delivery.

#### 2.1.3. Co-Delivering DOX with Other Anticancer Drugs

ZIFs have also been demonstrated to co-deliver DOX alongside other therapeutics [42,45,55]. Multidrug resistance is one of the main causes of chemotherapy failure in cancer, with the primary reason being the overexpression of active efflux transporters such as the P-glycoprotein [55]. Co-delivering drugs through a zeolitic framework holds the potential of overcoming multidrug resistance and increasing the targeting ability of the drugs. Zhang et al. utilized methoxy poly(ethylene glycol)-folate stabilized ZIF-8 to efficiently co-deliver verapamil hydrochloride as a P-glycoprotein inhibitor along with DOX hydrochloride. The multidrug delivery system demonstrated much safer and more effective therapeutic properties and can be used as a promising formulation in reversing the multidrug resistance for targeted cancer therapy [55]. Shen et al. co-delivered two drugs with distinct properties—the hydrophilic DOX with the hydrophobic near-infrared photosensitizer dye IR780. The combined effects of the two drugs not only significantly improved the pH-responsive drug release of ZIF-90, but also facilitated precise drug delivery to CD-44 overexpressed tumors [45]. Finally, Yan et al. reported a unique approach to the dual-drug delivery system by loading a photosensitizer (chlorin e6) and DOX with the ZIF-8 coating layer on E. coli via the biomimetic mineralization method. MOF-engineered bacteria preserved its tumor selectivity and exhibited strong therapeutic effects in both in vitro and in vivo experiments [42].

#### 2.1.4. Impact of MOF Size on DOX Delivery

Another important property to consider for MOFs in drug delivery is its size, which is commonly less than 200 nm to improve cellular uptake and blood-circulation time [87,88]. Yan et al. fabricated nanoscale size controllable and surface modifiable ZIF-8-poly (acrylic acid sodium salt) nanocomposites that ranged from 30 to 200 nm. These nanocomposites exhibited various crystallinity and pH sensitivity and retained their therapeutic efficacy when delivering DOX to cell lines and mice models [48]. Duan et al. proposed a one-pot, rapid, and completely aqueous approach to tune the size of DOX-loaded ZIFs. It was found that the 4T1 murine breast cancer cells tested were able to take up the DOX-loaded ZIFs in a size-dependent manner. In addition, an optimal size of 60 nm ZIF was shown to have longer blood circulation and over 50% higher tumor accumulation than its 130 nm counterpart [51]. Collectively the two studies showed that a biocompatible method to precisely control the size of ZIFs holds great potential in constructing multifunctional delivery systems for cancer theranostics and various other applications.

#### 2.1.5. Impact of PLNPs on DOX Release

Persistent luminescent nanoparticles (PLNPs) have been incorporated into the metal-organic framework of ZIF-8 to form multifunctional theranostic nanoplatforms that can improve the effectiveness and accuracy of tumor treatments [54,56]. Lv et al. constructed a ZIF-8 shell with PLNPs that possessed the dual functionalities of tumor imaging and pH-responsive drug delivery. The loading content of DOX on the nanoplatform reached a high percentage of 93.2%, and the release of DOX was greatly accelerated in the acidic environments created by tumor cells [54]. Similarly, Zhao et al. reported the anticancer properties of DOX-incorporated ZIF-8 with PLNPs. The theranostic platform can not only play a critical role in tumor imaging, but also showed anticancer drug loading capacity, acidity-responsive drug release behavior, and significant anti-tumor effect [56].

#### 2.1.6. Other Methods to Enhance Drug Release

Despite the advantages presented by ZIF-8 as a drug delivery system, it still possesses certain drawbacks, such as poor tumor targeting and short blood circulation time, that may reduce drug delivery efficiency [40]. To address this issue, a phosphorylcholine-based zwitterionic copolymer coated ZIF-8 nano-drug was developed. In the systemic circulation, the zwitterion can effectively extend blood circulation time to enhance tumor accumulation of the nanodrug. At the tumor site, the zwitterion can then rapidly convert to a positive charge, thereby enhancing tumor cellular uptake. This nanodrug is shown to have a 93.2% tumor inhibition rate on A549-bearing tumors with negligible side effects, suggesting great potential for this method of improving the efficiency of ZIF-8 [40].

In addition to the most common types of zeolites and ZIFs used, there are also some types, namely clinoptilolite and zeolite NaX, that have few previous studies on DOX release. For the first time, zinc-clinoptilolite/graphene oxide was fabricated and its cytocompatibility and drug loading capacity were investigated. The toxicity of the DOX-incorporated nanocomposite was also compared to that of pure DOX. The nanocomposite exhibited promising drug loading capacity and no toxic effects towards cells, especially below 16 mg/mL in concentration. In addition, the DOX-incorporated nanocomposite exhibited more cytotoxicity towards A549 lung tumor cells than free DOX [53]. Finally, magnetic zeolite NaX was combined with PLA/chitosan, Fe_3_O_4_, and/or ferrite with or without the presence of a magnetic field [50]. DOX loaded chitosan/PLA/NaX/ferrite with an external magnetic field after 7 days of treatment killed the most H1355 cancer cells (82% cell death) compared to all the groups. Overall, preliminary studies show that clinoptilolite and zeolite NaX also possess great potential in drug delivery and should be a topic of further investigation.

### 2.2. 5-Fluorouracil (5-FU)

5-fluorouracil (5-FU), an antimetabolite drug, is widely used in the treatment of cancer. 5-FU inhibits the activity of thymidylate synthase and incorporates its metabolites into RNA and DNA, thereby exerting anticancer effects [89]. Incorporating 5-FU into ZIF-7 modified with both metal ions and organic ligands showed a synergistic therapeutic effect in damaging the DNA and inhibiting the chemokine receptor 4 of esophageal squamous cancer cells [66]. In addition, Jiang et al. utilized pressure-induced amorphization to load a large amount of 5-FU into amorphous ZIF-8. Amorphous ZIF-8/5-FU was shown to have significant therapeutic efficacy in tumor-bearing mice due to less drug released during circulation, longer circulation time, and great biocompatibility [59]. Furthermore, Kulkarni et al. characterized 5-FU in the ZIF-8 framework using techniques such as FTIR, PXRD, Raman spectroscopy, EDX, and UV-NIR spectroscopy as well as morphological techniques such as SEM, TEM, and AFM [60].

Like those utilizing DOX, the ZIF studies incorporating 5-FU also harnessed the pH sensitivity of ZIF to produce a pH sensitive nanoplatform. Pandey et al. combined proteins, biopolymers, and ZIF-8 to construct a pH responsive nanoplatform for effective therapy against glioblastoma. In vitro cell line studies showed increased cancer cell cytotoxicity, which was further supported by the generation of cellular and surface reactive oxygen species by the nanocomposite [61]. Xiao et al. designed a novel ZIF-90@zinc oxide drug carrier that has a high 5-FU loading rate of 39%, which it will release in the acidic tumor microenvironment. Interestingly, the zinc oxide can decompose into Zn^2+^, which acts as an alternative therapeutic agent to overcome potential tumor resistance to 5-FU [64].

In addition to ZIFs, studies have also investigated how the unique properties of different types of zeolites affect loading capacity and release potential of 5-FU [62,63]. Vilaca et al. studied the drug delivery properties of FAU (zeolite NaY and zeolite nano NaY) and Linde Type L on colorectal cancer cell lines. In the first 10 min, in vitro drug release studies showed that 80–90% of 5-FU were released from the zeolites [62]. In addition, the differing pore sizes of various types of zeolites (FAU, BEA, MFI, LTA) were demonstrated to influence the loading capacity and release profile of 5-FU [63,67]. Sagir et al. found that 5-FU loaded magnetite-zeolite nanocomposites effectively inhibited the proliferation of gastric cancer cells lines through apoptotic mechanisms in vitro and may be a beneficial therapeutic agent against cancer [65]. Finally, Abd-Elsatar et al. showed that the release of 5-FU from zeolites (ZSM-5, zeolite A, FAU) are also pH dependent. The drug release occurred in two stages, and there was a significantly higher concentration of drugs in the more acidic media of gastric solution (pH 1.6) compared to a mildly acidic one (pH 5) [68]. Overall, 5-FU loaded zeolites hold as great a potential as 5-FU loaded ZIFs, and further animal studies should be conducted to determine its tumor inhibiting effects in vivo.

### 2.3. Curcumin

Curcumin, a natural phenolic drug extracted from turmeric, holds strong bioactive molecules known as curcuminoids to reduce cancers at the initial, promotion, and progression stages of tumor development [14,90]. Curcumin acts on cancers by blocking growth enzymes, modulating cellular progressions, and inhibiting lipid peroxidation and reactive oxygen species production [90]. Despite the promising anticancer effects of curcumin, the drug is poorly soluble in aqueous solutions, resulting in poor bioavailability that is somewhat mitigated by a very high dosage in oral formulations [14]. This traditional method of curcumin administration is not optimal; thus, a new route should be explored to enhance the drug’s efficacy. Zeolite may serve as a potential pharmaceutical carrier to increase the dissolution of curcumin as a therapeutic agent.

The surface properties and morphology of curcumin-loaded zeolite 5A was examined by Abadeh et al. using scanning electron microscopy (SEM), powder X-ray diffraction (XRD), differential scanning calorimetry (DSC), and UV-vis spectroscopy. These tests showed promising results by verifying the presence of curcumin in the zeolite framework, thereby supporting the potential use of the nanoplatform in targeted cancer therapeutics [14]. Curcumin-loaded nanoscale ZIF-8 (nZIF-8) and ZIF-8 were reported to have high drug encapsulation efficiency, good chemical stability, and fast drug release in the acidic tumor microenvironments. In addition, both nZIF-8 and ZIF-8 promoted cellular uptake of curcumin, which resulted in higher cytotoxicity towards HeLa cells [69,70]. Similar findings in antitumor efficacy were found in in vivo anticancer experiments of curcumin/nZIF-8 on mice [69]. The results indicate that curcumin-incorporated zeolites and ZIFs have great potential as efficient carriers for passive tumor therapy in future cancer treatments.

### 2.4. Cisplatin

Cisplatin is a chemotherapy drug that crosslinks with DNA’s purine bases to cause DNA damage and interfere with its repair mechanisms, thereby inducing apoptosis in cancer cells [91]. Cisplatin-loaded ZIF-90 with mitochondrial targeting was shown to have higher cellular uptake and less toxicity than cisplatin alone towards epithelial ovarian cancer cells. Incorporating cisplatin into ZIF-90 can also increase the specificity of drug release by producing a pH- and ATP-responsive nanoplatform [72]. In addition, the drug is often used in combination with other anticancer compounds to overcome tumor drug-resistance and reduce the inherent toxicity of the compound [91]. For example, cisplatin can be co-delivered with oleanolic acid to reverse multidrug resistance and induce apoptosis. Co-delivering the two drugs together in ZIFs yielded greater cancer cell death than the free drugs alone or mono delivery systems [71].

### 2.5. miR-34a

MicroRNAs (miRNAs) have become part of a promising class of nucleic acid drugs due to its vital role in miRNA modulation therapy. However, there are certain delivery challenges, mainly due to in vivo instability and low delivery efficiency, that impede the advancement of miRNA therapy [74]. Studies have incorporated miR-34a, a tumor-suppressing microRNA, into zeolites/ZIFs to enhance the tissue-specificity and safety of microRNA modulation therapy [73,74]. These novel nanoplatforms were successfully fabricated both with ZIF-8 and ZSM-5, thus demonstrating good biocompatibility in both ZIFs and zeolites. Release of the miR-34a-mimic@ZIF-8 complex decreased Bcl-2 expression at both mRNA and protein levels and promoted cellular apoptosis [74]. In vivo mouse model experiments also revealed miR-34a-mimic@ZIF-8 as a promising nanoplatform that can inhibit tumor growth via synergistic gene/chemodynamic therapy [74]. Incorporating miR-34a into ZSM-5 showed similarly promising results both in vitro and in vivo by inhibiting target oncogenes such as AEG-1 and SOX-9 [73]. Overall, miR-34a is a powerful candidate for cancer treatment, and incorporation of the miRNA into zeolites/ZIFs can mitigate the delivery challenges that miRNA therapy faces.

### 2.6. Miscellaneous Drugs

In addition to DOX, 5-FU, curcumin, cisplatin, and miR-34a, a variety of other drugs have been studied with zeolites/ZIFs acting as nanocarriers, with ZIF-8 being the most popular choice. Faraji Dizaji et al. loaded Paclitaxel into zeolite (ZSM-5) and MOFs (MIL-101 and ZIF-8) and saw that the MOFs had higher loading and more sustained release of the drug compared to their zeolite counterparts [75]. ZIF-8 can also be successfully used to deliver camptothecin [76], arsenic trioxide species [77], rapamycin [47], RNase A [78], gemcitabine [79], melittin [80], and lactate oxidase & Fe_3_O_4_ nanoparticles [81]. The nanoplatforms modified by camptothecin, arsenic trioxide, and rapamycin showed excellent pH-responsive hydrophobic anticancer drug delivery [47,76,77]. ZIF-8 is especially promising in delivering melittin, a hemolytic peptide whose conventional clinical applications are severely restricted due to its nonspecific hemolysis properties. The formation of a melittin@ZIF-8 complex can efficiently inhibit the hemolysis bioactivity of melittin until the nanocarrier reaches the desired location of a tumor microenvironment. This greatly increases the efficacy of the melittin@ZIF-8 complex towards the targeted induction of cancer cell apoptosis and tumor inhibition [80]. Zhou et al. simultaneously loaded lactate oxidase and Fe3O4 nanoparticles into ZIF-8 for a dual-modal therapeutic role. The combined effects of the two compounds is able to provide a simple, safe, and effective method to suppress rapid tumor growth and kill tumor cells [81].

In addition to ZIF-8, Zeolite Y (faujasite) has also been used as a popular carrier in anticancer drug delivery research. Amorim et al. investigated the suitability of α-cyano-4-hydroxycinnamic acid (CHC), an experimental anticancer drug, in zeolite NaY and zeolite NaA (LTA). CHC@zeolite exhibited up to 585 times the cytotoxic effects of the non-encapsulated drug, indicating its great potential in enhancing the effects of CHC [82]. Zeolite NaY was also successfully used to incorporate docetaxel, an anticancer drug, and protoporphyrin IX, a photosensitizer in a combined therapy using photodynamic therapy and chemotherapy [83]. Finally, zeolite Y was loaded with temozolomide (TMZ), a chemotherapeutic drug conventionally used to treat glioblastoma brain tumors. However, TMZ@zeolite Y did not have as strong of a cytotoxic effect as TMZ@mordenite, a natural zeolite [16]. Other natural zeolites that have been investigated are clinoptilolite, chabazite, and natrolite, which possess inherent cytotoxic properties and can reduce colorectal cancer Caco2 cell viability by 30, 40, and 60%, respectively. The toxicity of clinoptilolite and chabazite can be enhanced to 57 and 60%, respectively, with the binding and subsequent release of binase [84]. Clinoptilolite has also been modified with quercetin and quercetin dihydrate, both pharmaceutically active flavonoids. Although both drugs showed enhanced cytotoxicity, quercetin dihydrate@clinoptilolite showed greater cytotoxicity than quercetin@clinoptilolite [85]. Overall, drug-loaded natural zeolites also possess strong anticancer properties like their synthetic counterparts, and further research should be conducted to compare the difference in effectiveness of natural and synthetic zeolites in delivering various anticancer drugs.

## 3. Materials and Methods

This systematic review was conducted in accordance with the Preferred Reporting Items for Systematic Reviews and Meta-Analyses (PRISMA) guidelines (Appendix A). The following PICOS framework was used: Problem (P): efficacy of anticancer drugs through conventional delivery methods; intervention/indicator (I): the incorporation of the drug and compound into zeolites/ZIFs carriers; control (C): free anticancer drugs without incorporation into a zeolitic carrier; outcome (O): anticancer drug-incorporated zeolites/ZIFs can be used as alternative treatment options to enhance the efficacy of cancer treatment by mitigating the drawbacks of the drugs under conventional treatment options; and (S): in vitro and in vivo studies. The research question is: can zeolites/ZIFs enhance the therapeutic effects of anticancer drugs by acting as a drug carrier nanoplatform?

### 3.1. Literature Search Strategy

An exhaustive search of PubMed, Scopus, Embase, and Web of Sciences, was conducted, and all published studies were accumulated up to 25th June 2021. The grey literature search was conducted through ProQuest Dissertations & Theses Global and ProQuest ABI/INFORM global. There were no limitations set on the year or language of the publication. Across all databases and grey literature, free terms were searched with the search strategy: “zeolite AND (neoplasm OR neoplasia OR neoplastic disease OR tumor OR cancer OR carcinoma OR malignancy OR precancerous conditions)”. In addition, a controlled vocabulary search was conducted in PubMed (MeSH terms) and Embase (Emtree terms). The searches through the electronic databases were completed by J.H. with the help of Penn Dental Medicine Librarian Laurel Graham. The following search terms were used: MeSH terms: “zeolites”, “neoplasms”; Emtree Terms: “zeolite”, “neoplasm”. In the grey literature, the following filters were put in place to narrow down the number of searches: ProQuest Dissertations & Theses Global: biomedical engineering; ProQuest ABI/INFORM global: “Conference Papers & Proceedings” OR “Dissertations & Theses” OR “Other Sources” OR “Working Papers”.

### 3.2. Eligibility Criteria

Full-text studies that pertained to the therapeutic effect of zeolites/ZIFs as a drug delivery system on cancers were included in this systematic review. Studies that only reported the effect of pure zeolites/ZIFs on cancers were excluded from analysis. In addition, case reports, abstracts, notes, short communications, observational studies, and review articles/letters were excluded.

### 3.3. Screening and Selection

The studies accumulated were screened independently by two researchers (I.S. and J.H.) for titles and abstracts that were relevant to the subject and met the identified inclusion criteria. Any differences in options were discussed amongst the researchers until a consensus was reached. Next, the researchers found full-text articles of the studies and further assessed them for inclusion. Finally, the references of the selected articles were reviewed, and eligibility was determined based on the inclusion criteria. Any disagreements during this initial assessment process were resolved through consultation with review authors M.M.S. and F.O.

### 3.4. Data Extraction

Prior to data extraction, a protocol was jointly established by two of the authors (I.S. and J.H.). Data were then extracted from the selected full-text studies and compiled on an excel sheet. The two authors extracted data such as authors, publication year, objective, type of zeolite, type of cancer/tumor, drug/therapeutic loaded in zeolite, type of study, cell line, model organism, conclusion, and risk of bias (Table 2).

### 3.5. Assessment of Risk of Bias of Reviewed Papers

Two reviewers (Z.B. and F.O.) independently assessed the methodological quality of each included study through a risk of bias assessment based on previous studies [92,93]. Each study was assessed based on the following parameters: (I) description of sample size, (II) presence of a control group, (III) blinded assessment of the experimental outcome, (IV) adequately addressing outcome data, (V) standardized sample preparation, and (VI) inclusion of a conflict-of-interest statement. A score of 0 was assigned to a criterion for a study if it was clearly reported, a score of 1 was assigned to a criterion if it was vague or insufficiently reported, and a score of 2 was assigned to a criterion if the information was not present. The six criterion scores for each study were then summed to obtain a cumulative score. Articles that were at low risk of bias scored between 0–3, moderate risk of bias scored between 4–8, and high risk of bias scored between 9–12. Finally, any disagreements between the two authors during the evaluation were later discussed and a consensus was reached.

### 3.6. Inter-Rater Reliability (IRR)

An inter-rater reliability (IRR) test for the risk of bias assessment was performed using a kappa calculator on SPSS Statistics (IBM, Armonk, NY, USA) following the procedure outlined by McHugh [94]. Percent user agreement was calculated by taking the number of studies given the same risk of bias designations by both authors and dividing it by the total number of studies. To obtain the percent of data that are reliable, the kappa values were squared. From these percentages, a level of agreement was described for each parameter using [94].

## 4. Results

### 4.1. Search and Selection

A total of 1279 potentially relevant records were identified from the databases, grey literature, and reference search (Figure 1). After removing the duplicates, the titles and abstracts of 707 records were examined. Then, 631 studies were excluded because they did not meet the eligibility criteria, and 76 articles were assessed by full-text reading. Of the 76 studies retained for detailed review, 23 were not included because they did not meet the inclusion criteria. A total of 53 studies fulfilled all the selection criteria and were included in this review.

### 4.2. Risk of Bias Test of the Studies in the Systematic Review

The risk of bias data of the included studies were analyzed, and of the 53 articles, 33 were designated to have low risk of bias and 20 were designated to have moderate risk of bias by the two authors. Most papers analyzed were given a score of “2” by both authors in the “(III) blinded assessment of the experimental outcome” parameter for failing to provide any pertinent information (Table 3).

### 4.3. Inter-Rater Reliability Results

Results of the inter-rater reliability (IRR) test for each risk of bias criterion are shown in Table 4. All parameters showed at least 94% or higher in percent user agreement and the average percent user agreement across all five parameters was 97.8%. The average percent of data that are reliable, as determined by Cohen’s Kappa Test, is 81.53%, indicating an almost perfect level of reliability overall. A parameter that had a particularly weak level of agreement was whether the sample preparation was standardized. The authors’ disagreements regarding this parameter were explained by the variability in explanation provided by the articles. While the sample preparation method was explicitly based on a previous study in some articles, others vaguely mentioned that the samples were prepared using a standard method, thus leading to discrepancies in the initial analysis. However, all disagreements were resolved after discussion between the authors.

## 5. Limitations of the Study

The main limitation of this systematic review was that a quantitative evaluation through meta-analysis could not be performed due to the methodological heterogeneity among the evaluated studies. Since many of the included studies covered in this systematic review conducted in vitro experiments, it is important to note that there is a lack of widely accepted and clear criteria for assessing the risk of bias and quality of in vitro studies.

## 6. Conclusions

The available evidence collected through the present systematic review shows that zeolites/ZIFs hold great potential in delivering anticancer drugs in a targeted and controlled manner to tissues and organs. Specifically, in vitro cytotoxicity and in vivo tumor inhibition tests reveal that the nanoplatforms incorporated with conventional drugs such as DOX, 5-FU, curcumin, cisplatin, miR-34a, and others can successfully deliver anticancer therapeutics into the tumor microenvironment and enhance cancer cell uptake of the drugs. This specificity in targeting cancer cells is due to zeolite’s unique ability to degrade under acidic conditions, which is created by the tumor microenvironment. Although ZIF-8 is the most common nanocarrier used in the studies, other types of ZIFs and zeolites also showed promising results as effective drug carriers. Moreover, innovative surface modifications of zeolites increase effectiveness of anticancer drug delivery and produce some specific therapeutic characteristics based on the composition of the materials used.

Based on the findings of this systematic review, the authors recommend utilizing anticancer drug-incorporated zeolites/ZIFs as alternative treatment options to enhance the efficacy of cancer treatment by mitigating the drawbacks of the drugs under conventional treatment options. More in vivo studies need to be carried out to further support the therapeutic potential of zeolites as tumor-specific drug delivery systems. In addition, despite the promising preliminary results shown by the papers included in this study, types of cancers observed by the papers are somewhat limited. It would be valuable to extend the application of zeolites/ZIFs as a drug delivery system to other types of cancers, such as oral cancer, that have not yet been investigated.

## Figures and Tables

**Figure 1 molecules-26-06196-f001:**
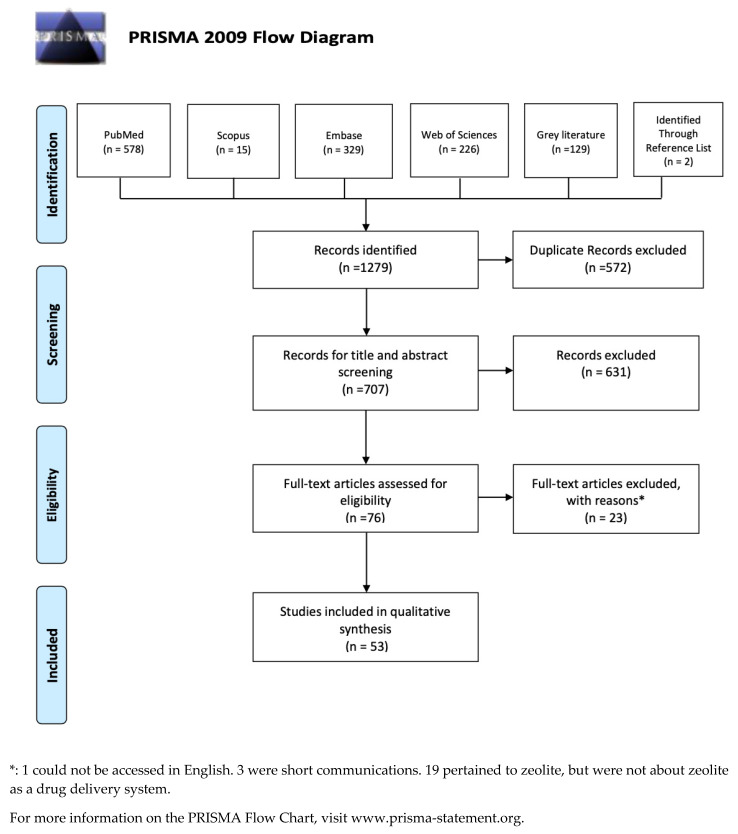
The Prisma Flow Diagram.

**Table 1 molecules-26-06196-t001:** Types of Zeolites and Zeolite-like Compounds Mentioned in This Systematic Review.

	Type	Name		Notes
Zeolite	Natural	Clinoptilolite		DDS
		Erionite		TOX
		Scolecite		TOX
		Offretite		TOX
		Chabazite		DDS
		Natrolite		DDS
		Mordenite		DDS
	Synthetic	Faujasite (FAU)	Zeolite X	DDS
			Zeolite Y	DDS
		Zeolite A/Linde Type A (LTA)	Zeolite 5A	DDS
			Zeolite NaA	DDS
		Linde Type L (LTL)		DDS
		ZSM-5		DDS
		Zeolite Beta (BEA)		DDS
Metal-Organic Framework (MOF)	Zeolitic Imidazolate Framework (ZIF)	ZIF-7		DDS
		ZIF-8		DDS
		ZIF-9		DDS
		ZIF-82		DDS
		ZIF-90		DDS
	MIL-101			DDS

Notes: DDS = has potential to serve as a drug delivery system; TOX = possesses inherent cytotoxic effects and therefore cannot serve as a drug delivery nanoplatform.

**Table 2 molecules-26-06196-t002:** Summary of the studies included in the systematic review.

Author, Year	Objective	Type of Zeolite	Type of Cacer/Tumor	Drug Loaded in Zeolite	Study Type	Cell Line	Model Organism	Conclusions
Abadeh et al. [14] (2020)	Used a combination of different techniques to investigate the adsorption properties of curcumin by zeolite type A for potential use as an anticancer drug carrier.	Zeolite 5A (LTA)	N/S	curcumin	in vitro	N/A	N/A	Confirmed the presence of curcumin in the zeolite 5A carrier and support the potential use of this porous material as drug carrier in targeted cancer treatments.
Martinho et al. [16] (2015)	The effect of zeolites and DDS was evaluated on the viability of glioblastoma cells in comparison with zeolites and temozolomide alone.	Zeolite Y (FAU); mordonite	Glioblastoma	temozolomide	in vitro; in vivo	U251	fertilized chicken eggs	Zeolite structures can be used effectively for sustained release of temozolomide in glioblastoma cells.
Yang et al. [21] (2018)	For the first time fabricated mesoporous ZSM-5 zeolites/chitosan core-shell nanodisks loaded with DOX as DDS against osteosarcoma.	ZSM-5	Osteosarcoma	DOX	in vitro; in vivo	MG63, hBMSCs	rats	The nanodisk drug carriers efficiently inhibited tumors with minor side effects, especially in cardiac toxicity.
Lei et al. [22] (2020)	Grew ZIF-8 on the surface of micelles to form a core-shell nanocomposite for the controlled release of DOX.	ZIF-8	Breast	DOX	in vitro	MCF-7	N/A	The core-shell nanocomposite could be a promising candidate for pH-responsive DDS in cancer therapy.
Li et al. [25] (2021)	Engineered silk sericin into ZIF-8 to overcome poor circulation stability and unexpected drug leakage into blood circulation, both issues that may limit the benefits of chemotherapy.	ZIF-8	Mammary	DOX	in vitro; in vivo	4T1	mice	The synthesized nanoplatform has tumor-specific biodegradability induced by the low pH environment, efficient drug uptake, and substantial tumor permeability effects.
Yan et al. [32] (2020)	Developed a proof of concept methodology for target-specific and pH-responsive delivery of DOX by ZIF-8.	ZIF-8	Breast	DOX	in vitro; in vivo	MDA-MB-231, MCF-10A	mice	The nanoplatform possesses inherent benefits of more precisely controlling the release of DOX in an acidic tumor microenvironment.
Tan et al. [37] (2021)	Reported the DOX-release and cytotoxic properties of DOX/MnO_2_@ZIF-8 as a chemophotothermal system.	ZIF-8	Lung	DOX, MnO_2_	in vitro; in vivo	LLC	mice	The as-prepared MnO_2_@ZIF-8 NPs with synergetic therapeutic effects by photothermal therapy and improved tumor microenvironment and as a pH-responsive nanocarrier for delivering the nonspecific anticancer drug DOX might be applied in the treatment of lung cancer.
Wu et al. [38] (2018)	Synthesized a biocompatible NIR and pH-responsive drug delivery nanoplatform based on ZIFs (PDA-PCM@ZIF-8/DOX) for in vivo cancer therapy.	ZIF-8	Liver	DOX	in vitro; in vivo	HepG2	mice	The biocompatible and biodegradable drug delivery platform based on ZIFs has shown great promise for future clinic cancer therapy.
Xie et al. [40] (2020)	Developed a phosphorylcholine-based zwitterionic copolymer coated ZIP-8 nanodrug, and the obtained nanodrug was prepared via a charge-conversional zwitterionic copolymer coating on DOX@ZIF-8 composites.	ZIF-8	Lung	DOX	in vitro; in vivo	A549	mice	This nanodrug is shown to have a 93.2% tumor inhibition rate on A549-bearing tumors with negligible side effects, suggesting great potential for this method of improving the efficiency of ZIF-8.
Yan et al. [42] (2020)	Loaded a photosensitizer (chlorin e6) and DOX with the ZIF-8 coating layer on E. coli (MG1655) via the biomimetic mineralization method. Subjected to in vitro cell viability assay and in vivo tumor treatment.	ZIF-8	Breast	DOX, chlorin e6	in vitro; in vivo	4T1	mice	MOF-engineered bacteria are powerful carriers for tumor-targeted therapeutic delivery.
Kang et al. [43] (2020)	Developed a convenient strategy and a proof-of-concept multifunctional composite for multimodal imaging and synergistic therapy of cancer using ZIF as a host matrix.	ZIF-8	Breast	DOX	in vitro; in vivo	4T1, L929, MCF-7, MCF-10a	mice	Demonstrated an applicable strategy to reveal the highly extendable capacity of ZIF-8 and integrated distinct multiple components in it to fabricate multifunctional composites for highly efficient cancer imaging and therapy.
Sharsheeva et al. [44] (2019)	Combined drug delivery nanocomposites with a semiconductor photocatalytic agent that would be capable of inducing a local pH gradient in response to external electromagnetic radiation.	ZIF-8	Neuroblastoma	DOX	in vitro	IMR-32	N/A	The system released DOX in a quantity sufficient for effectively suppressing IMR-32 neuroblastoma cells.
Shen et al. [45] (2020)	A robust trifunctional polymer coating strategy was proposed to address the major drawbacks of conventional ZIFs, while realizing synergistic chemo-photodynamic treatment by codelivering two therapeutics, chemotherapy drug DOX and near-infrared photosensitizer dye IR780.	ZIF-90	Cervical	DOX, IR780	in vitro; in vivo	HeLa	mice	The combined effects of DOX and IR780 not only significantly improved the pH-responsive drug release of ZIF-90, but also facilitated precise drug delivery to CD-44 overexpressed tumors.
Wen et al. [46] (2017)	Fabricated hollow mesoporous ZSM-5/chitosan ellipsoids loaded with DOX as pH-responsive drug delivery systems against osteosarcoma.	ZSM-5	Osteosarcoma	DOX	in vitro; in vivo	hBMSCs, MG63	rats	The HM-ZSM-5/CS/DOX ellipsoids as novel pH-responsive DDS can effectively treat osteosarcoma without systemic toxicity.
Xu et al. [47] (2020)	Aimed to provide a proof of concept for intentionally interfering cellular signaling pathway and autophagy for adjunct chemotherapy.	ZIF-8	Breast	Rapamycin (Rapa), DOX	in vitro; in vivo	MCF-7/ADR	mice	Rapa@ZIF-8 NPs provide a proof of concept for intentionally interfering cellular signaling pathway for adjunct chemotherapy.
Yan et al. [48] (2017)	Proposed a method of employing poly(acrylic acid sodium salt) (PAAS) nanospheres as a soft template to produce size controllable and surface modifiable ZIF 8-poly(acrylic acid sodium salt) nano-composites.	ZIF-8	Cervical	DOX	in vitro; in vivo	HeLa	mice	The nanocomposites exhibited various crystallinity and pH sensitivity, and retained their therapeutic efficacy when delivering DOX to cell lines and mice models.
Zheng et al. [49] (2020)	Presented strong upconversion luminescent biosafe cores derived from LTA and modification with targeted/therapeutic drugs for multimodal therapy, in which sonodynamic therapy (SDT) combined with photodynamic therapy (PDT) increases therapeutic efficiency especially in deep sites of tumor via producing cytoplasmic ROS and mitochondrial superoxide and photothermal therapy (PTT) enhances PDT effects via higher fluorescence resonance energy transfer (FRET) efficacy attributed to an increased temperature.	LTA	Melanoma	DOX	in vitro	B16-F0, 4T1, HBE, U937	N/A	The multimodal therapy allowed DOX@LTA zeolites to increase its therapeutic efficiency in the deep sites of tumors.
Abasian et al. [50] (2019)	Combined magnetic zeolite NaX with PLA/chitosan, Fe3O4, and/or ferrite with or without the presence of a magnetic field for sustained DOX release.	Zeolite NaX (FAU)	Lung	DOX	in vitro	H1355	N/A	DOX loaded chitosan/PLA/NaX/ferrite with an external magnetic field after 7 days of treatment killed the most H1355 cancer cells (82% cell death) compared to all of the groups.
Duan et al. [51] (2018)	Reported a one-pot, rapid, and completely aqueous approach that can precisely tune the size of drug-loaded MOF at room temperature.	ZIF-8 (amorphous)	Breast	DOX	in vitro; in vivo	4T1	mice	This size-controlled method helps to find the optimal size of MOF as a drug carrier and opens new possibilities to construct multifunctional delivery systems for cancer theranostics.
He et al. [52] (2019)	Examined the control of light and pH on the DOX hydrochloride-releasing properties of Au@ZIF-8.	ZIF-8	Cervical	DOX hydrochloride	in vitro	HeLa	N/A	Au@ZIF-8 with only 10 μM of DOX hydrochloride can result in 98% HeLa cell-killing activity after 30 min of light irradiation.
Khatamian et al. [53] (2016)	Synthesized Zn-Clinoptilolite/GO nanocomposite as an in vitro drug carrier system for DOX. Evaluated its drug loading capacity and studied its cytotoxicity using methyl thiazolyl tetrazolium (MTT) assay.	clinoptilolite	Lung	DOX	in vitro	A549	N/A	The prepared nanocomposite is cytocompatible and its high loading capacity and slow-release performance for DOX proved that it can be used as a drug carrier.
Lv et al. [54] (2019)	Reported the first core−shell multifunctional nanoplatform in the combination of persistent luminescent NPs and MOFs.	ZIF-8	Breast	DOX	in vitro; in vivo	4T1	mice	The loading content of DOX on the nanoplatform reached a high percentage of 93.2%, and the release of DOX was greatly accelerated in the acidic environments created by tumor cells.
Zhang et al. [55] (2017)	ZIF-8 is reported for the first time as the multidrug carrier to realizing the efficient co-delivery of verapamil hydrochloride (VER) as the P-glycoprotein inhibitor as well as DOX hydrochloride as an anticancer drug to overcome the MDR in addition to realize the active targeted ability for an efficient anticancer effect.	ZIF-8	Melanoma, Breast	DOX hydrochloride, Verapamil hydrochloride	in vitro; in vivo	B16F10, MCF-7	mice	The presented multidrug delivery system can be used as a promising efficient formulation in reversing the multidrug resistance for targeted cancer therapy.
Zhao et al. [56] (2019)	Reported the anti-cancer properties of DOX-incorporated persistent luminescent metal-organic framework (PLMOF).	ZIF-8	Breast	DOX	in vitro; in vivo	4T1	mice	The theranostic platform can not only play a critical role in tumor imaging, but also showed anticancer drug loading capacity, acidity-responsive drug release behavior, and significant anti-tumor effect.
Zhang et al. [57] (2019)	Explored the combined effects of pH and a NIR laser constructed ZIF-8 Janus NPs with lactobionic acid-gold nanorods on CT image-guided synergistic chemo-photothermal theranostics.	ZIF-8	Liver; Breast	DOX	in vitro; in vivo	HepG-2; MCF-7	mice	This dual-stimulation method had advantages in both cancer imaging and inhibited tumors in vivo by releasing pre-loaded DOX.
Jiang et al. [59] (2021)	Provided the dependable evidence that aZIFs could improve tumor therapeutic effect in vivo	ZIF-8 (amorphous)	Esophageal	5-FU	in vitro; in vivo	ECA-109, MCF-7	mice	aZIF-8 with favorable biocompatibility and long blood circulation is expected to be a promising nano-system for efficacious cancer therapy in vivo.
Kulkarni et al. [60] (2021)	The potential of developed nanoplatform against Neuroblastoma was assessed using a cell line studies along with in vivo toxicity studies to ascertain its safety for after in vivo administration in Wistar rats.	ZIF-8	Neuroblastoma	5-FU	in vitro; in vivo	IMR-32, SHSY-5Y	rats	Successfully optimized the size and yield of Lf-TC NPs and developed a potential nanoplatform for the multimodal therapy of Neuroblastoma by loading 5-FU inside the ZIF-8 framework.
Pandey et al. [61] (2020)	A novel and unique pH responsive nanoplatform have been developed for multimodal therapy of glioblastoma using protein, biopolymer and MOFs.	ZIF-8	Glioblastoma	5-FU, zinc	in vitro	U87MG, RAW264.7	N/A	The results suggest that the nanoplatform is promising for dual drug delivery mediated multimodal therapy of cancer.
Vilaca et al. [62] (2013)	Studied the drug delivery properties of FAU (zeolite NaY and zeolite nano NaY) and Linde Type L on colorectal cancer cell lines.	Zeolite NaY (FAU); zeolite nano NaY (FAU), LTL	Colorectal	5-FU	in vitro	HCT-15, RKO	N/A	Unloaded zeolites presented no toxicity to both cancer cells, while all DDS allowed an important potentiation of the 5-FU effect on the cell viability.
Vilaca et al. [63] (2017)	Studied the potential of several silica microporous structures as hosts for 5-FU as DDS for in vitro models of colorectal and breast cancers.	FAU, MFI, LTA	Breast, Colon	5-FU	in vitro	MDA-MB-468, HCT-15	N/A	The differing pore sizes of various types of zeolites were demonstrated to have an effect on the loading capacity and release profile of 5-FU.
Xiao et al. [64] (2020)	Designed a novel biodegradable treatment system based on ZIF-90.	ZIF-90	Cervical	5-FU; ZnO	in vitro; in vivo	HeLa	mice	The 5-FU-ZIF-90@ZnO core-shell NPs are a potential pH-controlled drug release system that can be applied to tumor treatment.
Sagir et al. [65] (2016)	Investigated the shapes of the particles, their size, drug loading and releasing capacity and biological activities in gastric cancer cell line AGS.	magnetite–zeolite nanocomposites (MZNC)	Gastric	5-FU	in vitro	AGS	N/A	5-FU loaded MZNC efficiently inhibit the proliferation of AGS cells in vitro through apoptotic mechanisms, and may be a beneficial agent against cancer, however further animal study is still required.
Cao et al. [66] (2020)	Proposed a structural reconstruction method to effectively explore and improve the biomedical application of ZIFs in esophageal squamous cell cancer theranostics.	ZIF-7	Esophageal	5-FU	in vitro; in vivo	K-150, MCF-10A	mice	Incorporating 5-FU into ZIF-7 modified with both metal ions and organic ligands showed a synergistic therapeutic effect in damaging the DNA and inhibiting the chemokine receptor 4 of esophageal squamous cancer cells.
Spanakis et al. [67] (2014)	Zeolite particles with different pore diameter and particle size were loaded with 5-FU. The loaded zeolites were characterized by means of SEM, XRD, DSC, XPS, N2 physisorption and FT-IR.	Zeolite NaX (FAU), BEA	N/S	5-FU	in vitro	N/A	N/A	Higher loading of 5-FU was observed for NaX-FAU than BEA.
Abd-Elsatar et al. [68] (2019)	Prepared three types of micronized zeolites and loaded them with 5-FU to be used as delivery systems for oral administration. Tested its efficacy via a cytotoxicity test.	ZSM-5, Zeolite A (LTA), Zeolite NaX (FAU)	Colon	5-FU	in vitro	CaCo-2	N/A	The synthesized zeolite frameworks are proposed to be of strong potential drug delivery vehicle for the treatment of gastrointestinal cancer.
Zheng et al. [69] (2015)	Developed a straightforward nanoprecipitating method to prepare water dispersible curcumin (CCM)-loaded nanoscale ZIF-8 NPs.	ZIF-8	Cervical	Curcumin	in vitro; in vivo	HeLa	mice	Both the in vitro and in vivo anticancer experiments indicate that CCM@nZIF-8 has much higher antitumor effect than free CCM and nZIF-8 might be used as the effective drug delivery system for the treatment of carcinoma.
Tiwari et al. [70] (2017)	Enlightened a novel approach of single step fabrication of curcumin@ZIF-8 as a drug carrier and its application as stimuli responsive drug delivery systems via external stimuli involving change in pH and in presence of biomimetic cell membrane like environment using liposomes and SDS micelles.	ZIF-8	Cervical	Curcumin	in vitro	HeLa	N/A	curcumin@ZIF-8 is an efficient drug carrier for passive tumor therapy in future for cancer treatments.
Chen et al. [71] (2020)	Constructed a cancer cell membrane-decorated ZIF hybrid nanoparticle (HP) to codeliver cisplatin and oleanolic acid (OLA).	ZIF NPs	Bladder	cisplatin; oleanolic acid (OLA)	in vitro; in vivo	SW780; NIH3T3	mice	HP/cisplatin/OLA could enhance apoptosis while reverse multidrug resistance in SW780 cells than free drugs alone or monodelivery systems, which might be a suitable DDS for co-delivery of different drugs with great promise.
Xing et al. [72] (2020)	Established the significance of the mitochondria-targeting carrier (ZIF-90) in the treatment of platinum-resistant ovarian cancer by a new therapeutic strategy.	ZIF-90	Ovarian	cisplatin	in vitro	A2780	N/A	The mitochondria-targeting ZIF-90@DDP with high drug loading could trigger responsive drug release in mitochondria of epithelial ovarian cancer cells, inhibit cisplatin-resistant epithelial ovarian cancer cells, and reverse drug resistance.
Salah et al. [73] (2019)	Developed an inorganic-organic hybrid vehicle for the systemic delivery of the tumor suppressor miR-34a. Investigate the efficiency of the delivered miR-34a in the treatment of HCC in vitro and in vivo.	ZSM-5	Liver	MiR-34a	in vitro; in vivo	HepG2	mice	Incorporating miR-34a into ZSM-5 showed promising results both in vitro and in vivo by inhibiting target oncogenes such as AEG-1 and SOX-9.
Zhao et al. [74] (2021)	Discovered the dual roles of ZIF-8 as nanocarriers for miRNA delivery and adjuvants for chemodynamic therapy.	ZIF-8	Breast	miR-34a mimic (miR-34a-m)	in vitro; in vivo	MDA-MB-231	mice	Demonstrated MOFs as a promising nanoplatform for efficient synergetic gene/chemodynamic therapy.
Faraji Dizaji et al. [75] (2020)	Various zeolites including hydrophilic Y zeolite, hydrophobic ZSM-5 zeolite and metal organic frameworks (MOFs) including MIL-101 and ZIF-8 were incorporated into the PLGA/chitosan nanofibers for controlled release of Paclitaxel anticancer drug against prostate cancer in vitro and in vivo.	zeolite Y (FAU), ZSM-5, MIL-101, ZIF-8	Prostate	paclitaxel	in vitro; in vivo	LNCaP	mice	The results confirmed a better performance of anticancer drug loaded-hydrophobic NMOFs loaded-nanofibers compared with zeolites and hydrophilic NMOF loaded-nanofibers for controlled release of anticancer drug and treatment of cancers.
Dong et al. [76] (2019)	Constructed a RGD (Arg-Gly-Asp) modified camptothecin@ZIF-8 (RGD@CPT@ZIF-8) as a novel metal-organic frameworks-based hydrophobic DDS for targeted and enhanced cancer treatment.	ZIF-8	Cervical	camptothecin	in vitro	HeLa	N/A	The nanoplatform exhibited the superior property of target to the cancer cells due to the function with RGD. The RGD@CPT@ZIF-8 nanoplatform has shown the enhanced cancer cell treatment due to the excellent pH-responsive hydrophobic anticancer drug delivery and intracellular ROS generation.
Ettlinger et al. [77] (2019)	Developed a pH-responsive nanocarrier of arsenic trioxide based on a metal–organic framework. Studied its drug release kinetics at different pH values and evaluate its cytotoxicity.	ZIF-8	ATRT	arsenic trioxide	in vitro	ATRT (BT12 and BT16)	N/A	Taking into account the low cytotoxicity of the drug loaded NPs on fibroblast and their cytotoxicity on the selected cancer cell lines, which was comparable to the free drug, ZIF-8 is a very promising candidate for drug delivery of arsenic trioxide.
Jia et al. [78] (2019)	ZIF-8 was employed as a carrier for the encapsulation and intracellular delivery of RNase A, aimed to achieve a rapid release of proteins in an acidic environment.	ZIF-8	Lung	RNase A	in vitro	A549, L02	N/A	ZIF-8 could be used as an effective carrier to deliver the therapeutic protein RNase A into the cytosol, which will be beneficial for improving the efficacy of cancer treatment.
Kamal et al. [79] (2021)	Reported the synthesis and use of nZIF-8 as a nanocarrier that is loaded with gemcitabine and surface-functionalized with the RGD homing peptide ligand to actively-target and specifically-deliver the chemotherapeutic agent to lung cancer cells.	ZIF-8 NPs	Lung	gemcitabine	in vitro	A549, MRC-5	N/A	Demonstrated a new one-pot strategy for realizing a surface-functionalized zeolitic imidazolate framework that actively targets cancer cells via an autonomous homing peptide system to deliver a chemotherapeutic payload effectively.
Li et al. [80] (2018)	Used various techniques, including a transcriptome analysis, to investigate ZIF-8 NPs loaded with melittin, a cytolytic peptide.	ZIF-8	Lung; cervical	melittin (MLT)	in vitro	A549, HeLa	N/A	There is great potential in using MOFs as a simple and efficient nanoplatform for delivering cytolytic peptides in cancer treatment.
Qin et al. [58] (2020)	Presented a facile strategy for constructing a biodegradable nanoparticle of MIP-stabilized fluorescent ZIF-8 for targeted imaging and GSH/pH dual stimulation drug release.	ZIF-8	Breast, Kidney, Colon	DOX	in vitro; in vivo	MCF-7, LoVo, 293T	mice	Because of the active targeting ability, good biocompatibility, tumor-sensitive biodegradability, and effective drug release performance, FZIF-8/DOX-MIPs can be widely used in tumor imaging and treatment.
Zhou et al. [81] (2020)	Hierarchical porous ZIF-8 is fabricated to simultaneously load lactate oxidase (LOD) and Fe3O4 NPs for tumor therapy.	ZIF-8	Breast	lactate oxidase (LOD), Fe3O4 NPs	in vitro; in vivo	4T1, MCF-7	mice	The combined effects of the two compounds is able to provide a simple, safe, and effective method to suppress rapid tumor growth and kill tumor cells.
Amorim et al. [82] (2012)	The effect of the zeolites and CHC-loaded zeolite drug-delivery systems were evaluated on HCT-15 human colon carcinoma cell viability.	FAU; Zeolite NaA (LTA)	Colon	α-cyano-4-hydroxycinnamic acid (CHC)	in vitro	HCT-15	N/A	Both zeolites alone revealed no toxicity to HCT-15 cancer cells. Importantly, CHC@zeolite led to an inhibition of cell viability up to 585-fold when compared to the nonencapsulated drug.
Kannen et al. [83] (2020)	Investigated the simultaneous detection of an anticancer drug and a photosensitizer administered in cancer cells using the zeolite matrix to assess their uptakes in cancer cells.	Zeolite NaY (FAU)	Prostate	docetaxel	in vitro	PC-3, PC-3-DR (docetaxel-resistant)	N/A	Indicated the efficacy of photodynamic therapy for docetaxel-resistant cancer cells.
Khojaewa et al. [84] (2019)	Searched for a biocompatible mineral carrier that allowed the safe delivery and long-term action of binase needed for treatment of ras-expressing malignances, especially colorectal cancer.	clinoptilolite; chabazite; natrolite	Colorectal	binase	in vitro	Caco2	N/A	The toxicity of clinoptilolite and chabazite can be enhanced to 57 and 60%, respectively, with the binding and subsequent release of binase.
Tomeckova et al. [85] (2012)	Modified clinoptilolite with active pharmaceutical compounds quercetin and quercetin dihydrate and studied their anticancer activities.	clinoptilolite	Leukemia, Cervical, Breast, Lung	Quercetin, quercetin dihydrate	in vitro	Jurkat, CEM, HeLa, MCF-7, A549 and MDA	N/A	Although both drugs showed enhanced cytotoxicity, quercetin dihydrate@clinoptilolite showed greater cytotoxicity than quercetin@clinoptilolite.

Abbreviations: ZIF = Zeolitic Imidazolate Framework; DDS = Drug Delivery System; LTA = Linde Type A; LTL = Zeolite Type L; 5-FU = 5-fluorouracil; DOX = doxorubicin; FAU = Faujasite; MOF = metal-organic framework; NPs = nanoparticles; MZNC = magnetite zeolite nanocomposite; N/S = Not Specified; N/A = Not Applicable.

**Table 3 molecules-26-06196-t003:** Risk of Bias Considering Aspects Reported in Section 3.

Study	(I) Sample Size	(II) Control Group	(III) Blinded Assessment of Outcome	(IV) Adequately Addressing Outcome Data	(V) Standardized Sample Prep	(VI) Conflict of Interest Statement	Risk of Bias
Abadeh et al. [14] (2020)	2	2	2	0	0	2	moderate
Martinho et al. [16] (2015)	0	0	2	0	0	2	moderate
Yang et al. [21] (2018)	0	0	2	0	0	2	moderate
Lei et al. [22] (2020)	0	0	2	0	0	0	low
Li et al. [25] (2021)	0	0	2	0	0	0	low
Yan et al. [32] (2020)	0	0	2	0	0	0	low
Tan et al. [37] (2021)	0	1	2	0	0	0	low
Wu et al. [38] (2018)	0	0	2	0	0	2	moderate
Xie et al. [40] (2020)	0	0	2	0	0	0	low
Yan et al. [42] (2020)	0	0	2	0	1	0	low
Kang et al. [43] (2020)	0	0	2	0	0	0	low
Sharsheeva et al. [44] (2019)	0	0	2	0	0	0	low
Shen et al. [45] (2020)	0	0	2	0	0	0	low
Wen et al. [46] (2017)	0	0	2	0	0	0	low
Xu et al. [47] (2020)	0	0	2	0	0	0	low
Yan et al. [48] (2017)	0	2	2	0	1	0	moderate
Zheng et al. [49] (2020)	1	0	2	0	0	0	low
Abasian et al. [50] (2019)	0	0	2	0	0	2	moderate
Duan et al. [51] (2018)	0	2	2	0	0	0	moderate
He et al. [52] (2019)	2	0	2	0	1	0	moderate
Khatamian et al. [53] (2016)	0	0	2	0	1	2	moderate
Lv et al. [54] (2019)	0	0	2	0	1	0	low
Zhang et al. [55] (2017)	0	0	2	0	0	0	low
Zhao et al. [56] (2019)	0	0	2	0	1	0	low
Zhang et al. [57] (2019)	0	0	2	0	1	0	low
Jiang et al. [59] (2021)	0	0	2	0	0	0	low
Kulkarni et al. [60] (2021)	0	0	2	0	0	2	moderate
Pandey et al. [61] (2020)	0	0	2	0	0	0	low
Vilaca et al. [62] (2013)	0	0	2	0	0	2	moderate
Vilaca et al. [63] (2017)	0	0	2	0	0	2	moderate
Xiao et al. [64] (2020)	0	0	2	0	0	0	low
Sagir et al. [65] (2016)	0	0	2	0	0	2	moderate
Cao et al. [66] (2020)	0	0	2	0	1	0	low
Spanakis et al. [67] (2014)	2	0	2	0	0	2	moderate
Abd-Elsatar et al. [68] (2019)	0	0	2	0	0	0	low
Zheng et al. [69] (2015)	0	0	2	0	1	0	low
Tiwari et al. [70] (2017)	0	0	2	0	1	0	low
Chen et al. [71] (2020)	0	2	2	0	0	0	moderate
Xing et al. [72] (2020)	0	0	2	0	0	0	low
Salah et al. [73] (2019)	0	0	2	0	0	0	low
Zhao et al. [74] (2021)	0	0	2	0	1	0	low
Faraji Dizaji et al. [75] (2020)	0	0	2	0	0	2	moderate
Dong et al. [76] (2019)	0	2	2	0	0	2	moderate
Ettlinger et al. [77] (2019)	0	2	2	0	1	0	moderate
Jia et al. [78] (2019)	0	0	2	0	0	0	low
Kamal et al. [79] (2021)	0	0	2	0	0	0	low
Li et al. [80] (2018)	1	0	2	0	0	0	low
Qin et al. [58] (2020)	0	0	2	0	0	0	low
Zhou et al. [81] (2020)	0	0	2	0	1	0	low
Amorim et al. [82] (2012)	0	0	2	0	0	0	low
Kannen et al. [83] (2020)	2	0	2	0	0	2	moderate
Khojaewa et al. [84] (2019)	0	0	2	0	1	0	low
Tomeckova et al. [85] (2012)	0	1	2	0	1	2	moderate

**Table 4 molecules-26-06196-t004:** IRR Values of Studies in the Systematic Review.

	% User Agreement	Kappa	% Data That Are Reliable (through Cohen’s Kappa Test)	Level of Agreement
(I) Sample Size	96.23%	0.784	61.47%	Moderate
(II) Control Group	96.23%	0.784	61.47%	Moderate
(III) Blinded Assessment of Outcome	100.00%	1	100%	Almost Perfect
(IV) Adequately Addressing Outcome Data	100.00%	1	100%	Almost Perfect
(V) Standardized Sample Prep	94.34%	0.814	66.26%	Strong
(VI) Conflict of Interest Statement	100.00%	1	100%	Almost Perfect

## Data Availability

The data are available from the corresponding authors upon reasonable request.

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
