# Peer review of "Effects of Zeolite as a Drug Delivery System on Cancer Therapy: A Systematic Review"

_molecules, 2021, doi:10.3390/molecules26206196_

Round 1
Reviewer 1 Report
In the manuscript, the Authors have reviewed the therapeutic effects of zeolites and ZIFs as carriers of anticancer drugs screening all published studies accumulated up to June 25th, 2021.
The work is properly executed and the conclusions are adequately supported by the data presented, taking into account that one of the authors (Pavelić, K) has published previously some reviews in the same field (most of all cited in the paper).
Therefore I pleased to recommend publication in Molecules, after some minor revisions.
- Line 44-46 are the same of line 52-54
- Figure of flow diagram (page 23) is too large and do not fit into page
- reference 88, Allendorf, M.D.; Bauer, C.A.; Bhakta, R.K.; Houk, R.J. T Luminescent Metal-Organic Frameworks. Chem. Soc. Rev. 2009, 38, 1330 - 1352
Author Response
Dear Reviewer,
Thank you very much for offering your suggestions for our systematic review paper titled "Effects of Zeolite as a Drug Delivery System on Cancer Therapy: A Systematic Review". We really appreciate your helpful feedback and valuable suggestions. Below is our response to your suggestions:
- Suggestion #1: Line 44-46 are the same of line 52-54
- Response: Line 44-46 has been deleted to remove the duplicate.
- Suggestion #2: Figure of flow diagram (page 23) is too large and do not fit into page
- Response: The flow diagram on page 23 has been adjusted to fit the page.
- Suggestion #3: reference 88, Allendorf, M.D.; Bauer, C.A.; Bhakta, R.K.; Houk, R.J. T Luminescent Metal-Organic Frameworks. Chem. Soc. Rev. 2009, 38, 1330 - 1352
- Response: A “T.” has been added to “Houk, R.J.” in reference 88.
Reviewer 2 Report
This study was to evaluate the potential therapeutic applications of zeo-lites/ZIFs as drug delivery systems delivering doxorubicin (DOX), 5-fluorouracil (5-FU), curcumin, cisplatin, and miR-34a. The manuscript could be considered for acceptance but not in its current form. Having said that following revisions are suggested. Comments: The abstract is descriptive and qualitative. It should be focused on the theme. The given list of keywords is superficial with broader terms. More specific terms should be used. Replace accordingly. The introduction is short. More literature should be added with recent and relevant literature. The novelty of the study should be clearly highlighted in the manuscript at the end of the introduction section. All sections should be critically discussed and compared with the previous reports. This will actually strengthen the manuscript and will highlight the significance of the work. The conclusion is superficial. Herein, I would like to see the major findings and how they are addressing the left behind research gaps and covering current challenges. Literature needs to be updated with care.Author Response
Dear Reviewer,
Thank you very much for offering your suggestions for our systematic review paper titled "Effects of Zeolite as a Drug Delivery System on Cancer Therapy: A Systematic Review". We really appreciate your helpful feedback and valuable suggestions. Below is our response to your suggestions:
- Suggestion #1: The abstract is descriptive and qualitative. It should be focused on the theme.
- Response: We edited the abstract and added more explanatory information to fit the theme of the systematic review.
- Suggestion #2: The given list of keywords is superficial with broader terms. More specific terms should be used. Replace accordingly.
- Response: We replaced certain broad keywords with more specific terms.
- Suggestion #3: The introduction is short. More literature should be added with recent and relevant literature. The novelty of the study should be clearly highlighted in the manuscript at the end of the introduction section.
- Response: We increased the length of the introduction by incorporating more specific examples from recent and relevant literature. We also expanded on the end of the conclusion by highlighting the novelty of the study.
- Suggestion #4: All sections should be critically discussed and compared with the previous reports. This will actually strengthen the manuscript and will highlight the significance of the work.
- Response: In our opinion, the discussion is already relatively lengthy compare to other parts of the paper. We tried to discuss the findings in different sessions and paid attention not to expand it too much with confusing information. The table 2 also gives very detailed information on the studies.
- Suggestion #5: The conclusion is superficial. Herein, I would like to see the major findings and how they are addressing the left behind research gaps and covering current challenges.
- Response: We worked on the conclusion. It has been revising to address the research gaps and potential areas of future research.
- Suggestion #6: Literature needs to be updated with care.
- Response: In-text citation numbers have been updated with care following the addition of new citations.
Reviewer 3 Report
The article "Effects of Zeolite as a Drug Delivery System on Cancer Therapy: A Systematic Review" shows an updated review of the possible application of zeolites in the transport and delivery of anticancer drugs.
Here are some suggestions for the document:
Review the content presentation format, Discussion appears first, then Materials and Methods, and then Results. In a traditional scheme I would expect Methods, Results and Discussion.
The introduction seems very well structured because it mentions the state of the art of the application of zeolites in cancer. However, the amount of information analyzed in the examples is somewhat limited. Possibly the authors can add a short section of "Future perspectives" where the future of zeolites in cancer is analyzed critically and from a suggestive point of view, in terms of research, but also of clinical trials and for commercial application.
Author Response
Dear Reviewer,
Thank you very much for offering your suggestions for our systematic review paper titled "Effects of Zeolite as a Drug Delivery System on Cancer Therapy: A Systematic Review". We really appreciate your helpful feedback and valuable suggestions. Below is our response to your suggestions:
- Suggestion #1: Review the content presentation format, Discussion appears first, then Materials and Methods, and then Results. In a traditional scheme I would expect Methods, Results and Discussion.
- Response: Following the guidelines of the journal, we ordered our manuscript as discussion, materials and methods, and results. We believe this is a special requirement of the journal.
- Suggestion #2: The introduction seems very well structured because it mentions the state of the art of the application of zeolites in cancer. However, the amount of information analyzed in the examples is somewhat limited. Possibly the authors can add a short section of "Future perspectives" where the future of zeolites in cancer is analyzed critically and from a suggestive point of view, in terms of research, but also of clinical trials and for commercial application.
- Response: We increased the length of the introduction by incorporating more specific examples from new citations. We also added a short section of “future perspectives” to both the end of the introduction and the conclusion.